# Whole-Genome Sequencing Snapshot of Clinically Relevant Carbapenem-Resistant Gram-Negative Bacteria from Wastewater in Serbia

**DOI:** 10.3390/antibiotics12020350

**Published:** 2023-02-08

**Authors:** Ivana Cirkovic, Bruno H. Muller, Ana Janjusevic, Patrick Mollon, Valérie Istier, Caroline Mirande-Meunier, Snezana Brkic

**Affiliations:** 1Institute of Microbiology and Immunology, Faculty of Medicine, University of Belgrade, 11000 Belgrade, Serbia; 2bioMérieux, R&D Microbiology, Sequencing Platform, F-69280 Marcy-l’Étoile, France; 3Institute of Virology, Vaccines and Sera “Torlak”, 11152 Belgrade, Serbia; 4bioMérieux, R&D Microbiology, Strain Collection & Characterisation, F-38390 La Balme Les Grottes, France; 5Institute for Laboratory Diagnostics “Konzilijum”, 11000 Belgrade, Serbia

**Keywords:** wastewater, antibiotic-resistant bacteria, carbapenems, colistin, whole-genome sequencing

## Abstract

Wastewater (WW) is considered a source of antibiotic-resistant bacteria with clinical relevance and may, thus, be important for their dissemination into the environment, especially in countries with poor WW treatment. To obtain an overview of the occurrence and characteristics of carbapenem-resistant Gram-negative bacteria (CR-GNB) in WW of Belgrade, we investigated samples from the four main sewer outlets prior to effluent into international rivers, the Sava and the Danube. Thirty-four CR-GNB isolates were selected for antimicrobial susceptibility testing (AST) and whole-genome sequencing (WGS). AST revealed that all isolates were multidrug-resistant. WGS showed that they belonged to eight different species and 25 different sequence types (STs), seven of which were new. ST101 *K. pneumoniae* (*bla_CTX-M-15_/bla_OXA-48_*) with novel plasmid p101_srb was the most frequent isolate, detected at nearly all the sampling sites. The most frequent resistance genes to aminoglycosides, quinolones, trimethroprim-sulfamethoxazole, tetracycline and fosfomycin were *aac*(6′)-Ib-cr (55.9%), *oqx*A (32.3%), *dfrA14* (47.1%), *sul*1 (52.9%), *tet*(A) (23.5%) and *fos*A (50%), respectively. Acquired resistance to colistin via chromosomal-mediated mechanisms was detected in *K. pneumoniae* (mutations in *mgrB* and *basRS*) and *P. aeruginosa* (mutation in *basRS*), while a plasmid-mediated mechanism was confirmed in the *E. cloacae* complex (*mcr-9.1* gene). The highest number of virulence genes (>300) was recorded in *P. aeruginosa* isolates. Further research is needed to systematically track the occurrence and distribution of these bacteria so as to mitigate their threat.

## 1. Introduction

Antimicrobial resistance (AMR) has been portrayed as one of the greatest threats to public health. AMR infections result in 700,000 deaths every year, but the global AMR-associated mortality is estimated to exceed 10 million lives per year in 2050 [1]. Hence, in 2017, the World Health Organisation (WHO) released a list of critical pathogens based on their impact on human health and the urgency for developing new antibiotics to treat resistant infections [2]. The group of Gram-negative bacteria, which includes carbapenem-resistant *Acinetobacter baumannii*, *Pseudomonas aeruginosa* and carbapenem-resistant and extended-spectrum beta-lactamases (ESBL)-producing *Enterobacterales*, was identified as a critical priority pathogen. Thus, few treatment options are available for infections caused by these bacteria [3] and such infections are associated with high morbidity and mortality rates worldwide [4].

Variations in AMR rates and patterns in different regions of the world have usually been associated with differing rates of antimicrobial consumption through overuse and misuse of antibiotics. Still, implementation of policies restricting antibiotic prescription has not decreased AMR rates as expected. Since 2015, antibiotic consumption in Serbia decreased by 18% [5,6]; however, AMR rates of all clinically relevant bacteria have increased (i.e., in the period 2015–2020, resistance rates to carbapenems in *Klebsiella pneumoniae* increased from 36% to 48%, in *Pseudomonas aeruginosa* from 47 to 69%, and *Acinetobacter* spp. from 91% to 97%) [7,8], demonstrating the importance of constraining other AMR-promoting factors.

Transmission of AMR can occur between humans, animals and the environment by means of a number of different routes. Wastewater is one of the primary pathways for clinically relevant antibiotic-resistant bacteria (ARB) to reach the environment [9]. Clinical wastewater, such as wastewater from healthcare institutions and nursing homes, are hotspots for ARB [9,10]. However, more than 80% of the total antibiotic consumption in the human sector is prescribed in the community in Europe [11] and these AMR indicators are also present in municipal wastewater. Therefore, testing wastewater for the presence and diversity of ARB could provide an indication of where and to what extent clinically relevant pathogens are released into the environment [12]. After discharge into the environment, ARB may be highly prevalent in soil, plants and surface water, and may, thus, pose a risk for the colonization of humans, pets, livestock or contamination of food and food products, and further tremendously impact public health [13].

In Serbia, based on the 2017 data, less than 13% of collected municipal and clinical wastewater was treated before discharge to receiving waters (e.g., rivers) [14]. Belgrade, the capital of Serbia with 1,700,000 inhabitants, does not have any wastewater treatment facility, and although Belgrade has quite a developed sewer system, most of the sewer outlets are submerged in the final recipients, the Sava and the Danube international rivers. The Danube is the second-longest river in Europe, which connects ten European countries, running through their territories or constituting a border [15]. Its drainage basin extends into nine more countries. The Danube passes through four capital cities, more than any other river in the world, and, additionally, five more capital cities lie in the Danube’s basin. The Sava is the longest and largest tributary of the Danube by discharge. It flows through five European countries, feeding into the Danube in Belgrade.

The impact of untreated wastewater discharge on the water quality of the Danube River was previously demonstrated [16]. In the river stretch through the central part of Serbia, all midstream samples were critically polluted based on *Escherichia coli* numbers (main faecal indicator), and the highest level of faecal pollution was recorded downstream of Belgrade. Of note is that ARB from the urine origin, in the case of human/animal urinary tract infections, are additionally present in wastewater and its recipients. Therefore, knowing that wastewater from Belgrade significantly deteriorates the microbiological water quality of the Sava and the Danube, the need to determine the presence of ARB is of critically significance for public health at this highly impacted stretch of the international rivers was identified.

Our study aimed to evaluate the diversity of carbapenem-resistant Gram-negative bacteria of clinical significance in wastewater prior to outlet in two international rivers, the Sava and the Danube.

## 2. Results

### 2.1. Distribution of Clinically Relevant ARB Isolated from Wastewater

Overall, 16 wastewater samples were collected and investigated within this study and 34 target antibiotic-resistant Gram-negative bacteria were isolated: *Klebsiella pneumoniae* (*n* = 12), *Klebsiella oxytoca* (*n* = 3), *Enterobacter cloacae* complex (*n* = 9), *Escherichia coli* (*n* = 3), *Serratia marcescens* (*n* = 2), *Citrobacter freundii* (*n* = 1), *Acinetobacter baumannii* (*n* = 2) and *Pseudomonas aeruginosa* (*n* = 2). Distribution of isolated target ARB depending on the sampling site and sampling campaign is presented in Table 1.

The strains of *K. pneumoniae* and *E. cloacae* complex were isolated from all sampling sites (the main sewer outlets of Belgrade) and during the whole study period. The diversity of the target ARB strains was present both in S1 and S2, which receive municipal wastewater with or without hospitals wastewater contribution, respectively.

### 2.2. Antimicrobial Resistance Pattern of Wastewater Isolates

An overview of the antimicrobial resistance of the investigated target bacteria is presented in Figure 1. MICs and interpretation of antimicrobial susceptibility testing for each isolate are listed in Appendix A.

*K. pneumoniae* isolates were resistant to antimicrobials of different classes (Figure 1a). Interestingly, phenotypic resistance to carbapenems was not uniform, i.e., 100% to ertapenem, 41.7% to imipenem and 75% meropenem. Similar findings were found for other antimicrobial classes (amynoglycosides and fluoroquinolones) and in others isolated ARB.

In contrast to isolates of *K. pneumoniae*, the *K. oxytoca* and *E. coli* isolates exhibited higher susceptibility levels, while *S. marcescens* and *C. freundii* isolates higher resistance levels to almost all tested antimicrobials (Figure 1a–c). Similar to the findings for *K. pneumoniae*, almost all *E. cloacae* complex isolates showed resistance to tested antimicrobial agents (Figure 1b).

Susceptibility to the newly approved antimicrobial combination ceftazidime-avibactam was 91.7% in *K. pneumoniae*, 100% in *K. oxytoca* and *C. freundii*, 66.7% in *E. cloacae* complex and *E. coli* and 50% in *S. marcescens*. On the other hand, seven (58.3%) isolates of *K. pneumoniae* and one (11.1%) isolate of *E. cloacae* complex showed resistance to last resort antibiotic colistin (Appendix A). *A. baumannii* and *P. aeruginosa* isolates were resistant to all tested antimicrobial agents with EUCAST clinical breakpoints, except colistin and amikacin, respectively (Figure 1d, Appendix A).

Finally, antimicrobial susceptibility testing revealed that all target ARB isolates described in this study were multidrug-resistant (MDR) [17].

### 2.3. Characterisation of β-lactamase Genes (ESBL and Carbapenemase Genes)

Among 34 sequenced target Gram-negative isolates, a high diversity of β-lactamase genes was detected (Table 2).

The majority of *Enterobacterales* isolates (70%) harbored more than one ESBL gene, in different combinations, which included *bla*_CTX-M-15_ and various types of *bla*_SHV_, *bla*_TEM_, *bla*_OXA_ and *bla*_OXY_ genes. The predominant ESBL gene in all *K. pneumoniae* isolates was *bla*_CTX-M-15_, followed by *bla*_OXA-1_ (83.3%). Genes for the SHV-212 type of enzyme were detected exclusively in *K. pneumoniae* isolates that belong to ST101 (58.3%). All isolates of *K. pneumoniae* harbored carbapenemase *bla*_OXA-48_ gene, with one isolate co-harboring *bla*_OXA-48_ and *bla*_NDM-1_. Besides β-lactamase genes, all *K. pneumoniae* isolates harbored the point mutation in the *omp*K36 locus, which can confer resistance to carbapenems and other β-lactams through reduced permeability. Out of three *K. oxytoca* isolates, two isolates carried *bla*_GES-5_ carbapenemase gene in combination with different ESBLs from OXA and OXY families and one isolate carried *bla*_OXA-48_ with *bla*_OXY_ ESBL gene. In *E. cloacae* complex, ESBL genes were found as follows: *bla*_OXA-1_ (55.5%), *bla*_TEM-1_ (44.4%), *bla*_CTX-M-15_ (22.2%), and other types of ESBLs were found in single isolates. These ESBLs were in combination with metallo-beta-lactamase, *bla*_NDM-1_ in 44.4% isolates and *bla*_IMI-2_ gene in one isolate of *E. cloacae* complex. All three isolates of *E. coli* were ESBL and carbapenemase producers. Two isolates were positive for the *bla*_NDM-1_ carbapenemase gene, and one isolate harbored the *bla*_OXA-48_ gene. *C. freundii* was carbapenemase negative, but ESBL positive, with the presence of *bla*_CTX-M-162_ and *bla*_TEM-1_ genes. Both *S. marcescens* isolates were positive for ESBL and carbapenemase genes, one with *bla*_SHV-2a_ and *bla*_GES-5_, the other with *bla*_TEM-1_ and *bla*_NDM-1_ genes. In *A. baumannii* and *P. aeruginosa* isolates, different cephalosporinases from ADC and PDC families were confirmed. One isolate of *P. aeruginosa* harbored the *bla*_PER-1_ ESBL gene, while the other was positive for *bla*_OXA-395_ chromosomal beta-lactamase. Both *A. baumannii* isolates carried class D carbapenemase (oxacillinase): *bla*_OXA-23_/*bla*_OXA-66_ and *bla*_OXA-66_/*bla*_OXA-72_, respectively.

### 2.4. Resistome Other Than β-lactamase Genes

Antimicrobial resistance genes (ARG) besides genes conferring resistance to beta-lactams are presented in Table 2.

Aminoglycoside modifying enzymes (AME) genes were detected in almost all the target ARB isolates (97.1%). Whole-genome sequencing (WGS) analysis showed heterogeneous AME types from *aac*-, *aad*-, *aph*-, *ant*- and *aba*- families, including the *armA* gene, which confers a high level of aminoglycoside resistance. The most frequent AME genes were *aac(6′)-Ib-cr*, *aph (6)-Id*, *aac (3)-IIe* and *aad A2*.

Genes conferring resistance to fluoroquinolones, *oqxA* and *oqxB*, were detected in all *K. pneumoniae* isolates, including *aac(6′)-Ib-cr* in 83.3% isolates, which confers resistance to both aminoglycosides and fluoroquinolones. In the *E. cloacae* complex, the *qnr* gene was found in 66.7% of isolates, as well as in one *E. coli* isolate. Resistance to fosfomycin, mediated by the *fosA* gene, was detected in all *K. pneumoniae* and *P. aeruginosa* isolates, and in 88.9% of *E. cloacae* complex isolates.

Tetracycline-resistance gene *tetD* was found in 50% of *K. pneumoniae* and in 33.3% of *E. coli* isolates, *tetA* in 41.7% of *K. pneumoniae* and in 33.3% of *E. cloacae* complex isolates, while *tetB* gene was confirmed in all *A. baumannii* and one *E. coli* isolates. Genes conferring resistance to sulfonamides, *sul1* and *sul2*, were found in 52.9% and 32.3% of target ARB isolates, respectively. Additionally, seven (20.6%) ARB isolates harbored both genes. The most common trimethoprim resistance gene was *dfrA14*, found in 66.7% *K. pneumoniae*, 66.7% of *E. cloacae* complexes and 33.3% of *E. coli* isolates. The second most common was *dfrA12*, detected in different species, but also in combination with *dfrA14* in 22.2% of *E. cloacae* complex isolates. Phenicol resistance was confirmed in all species, except *K. oxytoca* and *C. freundii*, with the *catB3* gene as the most common (52.9%). Finally, plasmid mediated colistin resistance was confirmed in one *E. cloacae* complex isolate (BS11105), which harbored the *mcr-9.1* gene. In one *K. pneumoniae* (BS11131) and two *P. aeruginosa* (BS11135; BS11136) isolates, mutations in the *basS/basR* two-component system were detected. Additionally, six of the seven ST101 *K. pneumoniae* isolates had a missense mutation (TGC->AGC) in the *mgrB* gene, leading to substitution C28S in the MgrB regulator (Appendix A).

### 2.5. Plasmidome and Virulome

In-silico plasmid detection confirmed 28 different plasmids in analyzed *Enterobacterales* (Appendix A).

The majority of *K. pneumoniae* isolates (91.7%) had four or more different plasmid replicon types. The most common was Col440II (75%), followed by IncFIA(HI1) (58.3%), IncR (58.3%), IncL (50%), IncFIB (pKPHS1) (50%) and others. As expected, ST101 *K. pneumoniae* shared similar plasmid profiles, all harboring Col440II, IncFIA(HI1) and IncR, among other replicon types. These replicons were part of plasmid contigs in 86.7% of ST101 isolates. Plasmid p101_srb (accession number: NZ_MN218814.1) contained IncFIA(HI1) and IncR replicons, and plasmid pKp_Goe_641.3 (accession number: NZ_CP018738.1) contained the Col440II replicon. Plasmids detected in the *E. cloacae* complex showed the highest diversity (20 replicon types in nine isolates). In *K. oxytoca*, GES-5-positive isolates shared a similar plasmid profile, whereas one OXA-48 positive isolate had IncL and IncFII(Yp) replicon. Each *E. coli* isolate had a unique plasmid profile, with replicons from the Inc group, as well as *C. freundii* and *S. marcescens*. Additionally, inter-species detection of plasmid replicons was observed.

Virulome analysis demonstrated a plethora of virulence genes (Appendix A) in 31 strains. However, in strains of *C. freundii* and *S. marcescens*, no virulence genes were detected. The most diverse virulome showed isolates of *P. aeruginosa*, with more than 300 genes. The vast majority were antiphagocytosis virulence genes (*alg44*, *alg8*, *algA-Z*), type II secretion system—xcp secretion system (*xcpP-Z*), type III secretion system (*pcr1-4*, *pcrD-V*, *pscB-U*), type VI secretion system (*tagF1-T*, *tse1-3*, *tssA1-M1*), genes for iron acquisition and pigment (*pvcA-D*, *pvdA-S*) and adherence and motility genes (*fimT-V*, *flgA-N*, *fliA-S*). *A. baumannii* isolates shared the same virulome profile with 29 virulence determinants: efflux pump (*adeF-H*), acinetobactin (*barB*, *basA-J*, *bauA-F*), and csu fimbriae (*csuA/B*, *csuB*, *csuD*). *K. pneumoniae* isolates had a quite similar distribution of virulence genes, all harboring AcrAB efflux pump, adherence genes (*fim*, *mrk*), genes for iron acquisition (*ent*, *fep*, *fyu*, *irp*, *ybt*) and secretion system T6SS-I and T6SS-III (*clpV/tssH*, *dotU/tssL*, *hcp/tssD*, *icmF/tssM*, *impA/tssA*, *tssF*, *tssG*, *vasE/tssK*, *vgrG/tssI*, *vipA/tssB*, *vipB/tssC*). In *E. coli*, 90 virulence genes were found, with one isolate that belonged to sequence type ST131, and harbored toxins genes (*hlyA-alpha*, *hlyB*, *hlyD*) and invasion factors (*aslA*, *kpsC*, *kpsE*, *kpsMII*, *kpsS*, *kpsU*, *ompA*, *ompT*) among other virulence determinants.

### 2.6. Genomic Epidemiology and Phylogenetic Relatedness

The 12 *K. pneumoniae* isolates resolved into six sequence types (STs), of which one isolate represented a new ST (ST6273). The most frequent was ST101 (*n* = 7, 58.3%), whereas other STs represented singletons: ST15, ST16, ST29, ST437 and ST6273. Among ST101 we detected two clusters (Figure 2a), genetically close (99.55% similarity), sharing the same resistome and the same plasmid replicons but not strictly the same virulome (Appendix A). Cluster 1 grouped two ST101 strains (BS11119 and BS11134) isolated from the same sampling site (S2), but different sampling campaigns (SC1-May and SC4-August). Cluster 2 grouped three isolates (BS11125, BS11126, BS11128), two from the same sampling site and sampling campaign (S1, SC3), and a third from the same sampling campaign (SC3), but from a different sampling site (S3), placed downstream from S1. Regarding virulome differences, compared to the other isolates of cluster 1 and 2, respectively, BS11119 had an additional virulence factor (*vipB/tssC*) belonging to the type VI secretion system (T6SS), and in BS11125, the *mrk* gene cluster (*mrkABCDF*) and regulatory genes (*mrkI*, *mrkJ*) were missing. Compared to isolates from both clusters, other ST101 (BS11122; BS11120) were more distant (<99% similarity) and had many differences in resistome (Appendix A). A minimum spanning tree (MST) displaying the *K. pneumoniae* isolates is presented in Figure 3 to show the allelic differences.

The nine *E. cloacae* complex isolates were resolved into nine STs: ST23, ST32, ST114, ST136, ST171, ST364, and three newly assigned STs: ST2006, ST2007, ST2008. These isolates showed no clustering and were genetically unrelated (Figure 2b). An MST displaying the *E. cloacae* complex isolates is presented in Figure 4, showing at least 2200 allelic differences between each sample.

One *K. oxytoca* isolate belonged to ST108, while the other two were a new sequence type (ST427). Both ST427 isolates, originating from different sampling sites and sampling campaigns (S1/SC2 and S3/SC3), were closely related (99.50% similarity), sharing the same resistome, but not the same virulome (*traT* gene only found in BS11117; no virulome feature in BS11118) (Figure 2c, Appendix A).

All isolates of *E. coli*, *C. freundii*, *A. baumannii* and *P. aeruginosa* represented singletons (Table 2). To date, no MLST scheme has been described for *S. marcescens*.

## 3. Discussion

The looming post-antibiotics era calls for urgent epidemiological measures for the containment of ARB linking all settings under the One-Health umbrella. However, the important role of wastewater and its contribution to the transmission of AMR in the environment is often not addressed or neglected [9,10,12,13]. The Belgrade sewer system consists of combined and separate sewers with a total length of over 1500 km, to which 1.7 million inhabitants are connected. Works on upgrading the sewer system and the introduction of wastewater treatment plants is ongoing; however, currently, sewage is still discharged into the rivers the Sava and the Danube without any treatment [14,18]. To our knowledge, this is the first comprehensive study to evaluate presence of ARB in the Belgrade sewer system prior to outlet to recipients and thereafter AMR pollution of two international rivers in their lower drainage basin, providing important additions to the detailed insight into the extent of the issue on a European level.

*K. pneumoniae* isolates from our study represents critical pathogens according to the WHO priority list, due to their MDR profile with resistance to third-generation cephalosporins and carbapenems [2]. We confirmed ESBL-positive isolates in samples from all sampling sites (S1–S4) and from all sampling campaigns. All isolates produced CTX-M-15, as well as other clinically relevant ESBLs from SHV, TEM and OXA families. CTX-Ms, usually located on epidemic resistance plasmids of incompatibility group F (IncF) [19], were detected in all *K. pneumoniae* isolates in our study. This contributes to the global spreading of CTX-M-15, representing the most widely distributed ESBLs, responsible for nosocomial outbreaks and community-onset infections [19,20].

An OXA enzyme with an ESBL phenotype, OXA-1, was the second most common ESBL among wastewater isolates in the study, found in 91.7% *K. pneumoniae.* On a global level, *bla*_OXA-1_ is in the top 15 ARGs from the environmental samples [20]. This class D enzyme is often found with other ESBLs in clinical isolates as well. Palmieri et al. detected *bla*_CTX-M-15_ and *bla*_OXA-1_ genes in the vast majority of clinical carbapenem- and colistin-resistant *K. pneumoniae* from Serbia [21]. Similar findings were described in the Croatian study of OXA-48 producing hospital isolates of *K. pneumoniae* [22]. Another specific result from our study relates to the detection of *bla*_SHV-212_ only among isolates that belonged to ST101. Overall data about this SHV variant are scarce, but it was described in a *K. pneumoniae* isolate from a Dutch healthcare worker [23].

The most concerning result was the detection of Ambler’s class D carbapenemase OXA-48 in all *K. pneumoniae* isolates. Detection of carbapenemase-positive *K. pneumoniae* was a consistent finding in the study, confirmed in all sampling sites and sampling campaigns. OXA-48 represents the most frequent carbapenemase from the environmental samples in Europe [20], and the most widespread carbapenemase among *Enterobacterales* globally [24]. The genetic environment of *bla*_OXA-48_ gene is well-described, including different mobile genetic elements, such as transposon Tn*1999* and pOXA-48a-like IncL (IncL/M) plasmid [24]. The IncL plasmid was confirmed in 50% of *K. pneumoniae* in this study, including one ST15 and one ST101 isolate. Another 50% of isolates, all belonging to ST101, harbored plasmid p101_srb with IncFIA(HI1) and IncR replicons. p101_srb was firstly described by Palmieri et al. in an ST101 *K. pneumoniae* clinical isolate from Serbia, encoding OXA-48, the CTX-M-15 ESBL and several other AMR genes, and conferring an MDR phenotype [21]. This finding strongly suggests the clonal expansion of the high-risk clone ST101, and dissemination from medical settings into the environment in Serbia.

Additionally, one ST101 isolate (BS11120) harbored genes for both OXA-48 and NDM-1 enzymes. Serbia is considered endemic for NDM carbapenemase [25]; thus, the co-occurrence of *bla*_OXA-48_ and *bla*_NDM_ genes could be expected, and it has been described in ST101 *K. pneumoniae* hospital isolates from Serbia [26].

MLST analysis revealed six STs among *K. pneumoniae* isolates, including a novel one—ST6273. The most prevalent was ST101, and the phylogenetic analysis of ST101 isolates showed two clusters. Cluster 1 grouped isolates (*n* = 2) from the same sewer outlet (S2, with no hospital input), but from different sampling campaigns (S1 and S4), indicating the persistence of this clone over time. Cluster 2 grouped isolates (*n* = 3) from the same sampling campaign (SC3), of which two were from sampling site S1, and one from the sewer outlet placed downstream (S3), indicated probable spreading through the ecosystem. These five isolates shared the same resistome and had the same plasmid replicons (Col440II, IncFIA(HI1), IncFIB(pKPHS1), IncR), but not strictly the same virulome. Two additional ST101 isolates were more distant compared to isolates from clusters, with differences in resistome. Nevertheless, all ST101 isolates showed the presence of virulence factors significant for hypervirulent *K. pneumoniae* strains, such as siderophores (enterobactin, yersiniabactin, salmochelin, aerobactin) and type 3 fimbriae, and can cause serious infections, such as liver and lung abscesses, in otherwise healthy individuals [27]. Thus, the dissemination of these ST101 in the environment in Serbia may present a worrisome public health issue. Additionally, we detected one ST437 isolate from sampling site S2, which has no hospital effluents. This clone was previously described among clinical isolates from Serbia, as a part of a single monophyletic subclade within the CG258 [21].

Multiple studies described *K. oxytoca* isolates from wastewater [28,29,30,31]. Among *K. oxytoca* isolates from our study, we confirmed carbapenemase gene *bla*_OXA-48_ in one isolate, belonging to ST108. This clone was found among hospital isolates in Australia [32], but overall data about it are scarce. The reason could be the limitation of phenotypic identification methods, which cannot distinguish members of *K. oxytoca* complex. In accordance with that, ST108 was described in *Klebsiella michiganensis* clinical isolates [33]. Interestingly, we observed phylogenetic relatedness among two other *K. oxytoca* wastewater isolates belonging to novel ST uncovered in this study, ST427, isolated from different sampling sites and sampling campaigns. They shared the same resistome, and formed a cluster with 99.50% similarity. One of these isolates possesses virulence gene *traT*, responsible for complement resistance.

*E. cloacae* complex was isolated from all sampling sites and sampling campaigns in our study. These isolates showed high diversity in terms of carbapenemase production, plasmid content, virulence genes and STs. Among carbapenemase-positive isolates, the *bla*_NDM-1_ gene was the most frequent. This gene is the most reported one in the environment in the past 30 years [20], whereas the hospital wastewaters are considered as the major source of *bla*_NDM_ variants, especially in India, an endemic area [34]. Serbia is considered endemic for the NDM carabapenemase as well [25], but *bla*_NDM_ was not confirmed in a previous study of environmental water in Belgrade, Serbia [35]. On the other hand, among clinical isolates of *Enterobacter* spp. from global surveillance programmes, *bla*_NDM_ was predominantly found among isolates from the Balkans, India and Vietnam [36]. Among *Enterobacter* spp. community isolates in Belgrade, Serbia, *bla*_NDM_ was the most prevalent carbapenemase gene [37]. In our study, all *E. cloacae* complexes belonged to different STs, and the heterogeneity of STs was demonstrated through discovery of three new STs among nine isolates. This is in accordance with the results of other studies, which have shown that spreading of *bla*_NDM-1_ was not related to a specific clone [25].

Among *E. coli* isolates, we detected ST131 and ST155 which are considered high-risk clones with zoonotic potential and disperse antibiotic resistance on a global scale. Extraintestinal MDR *E. coli* ST131 is a worldwide pandemic pathogen and a major cause of urinary tract infections, bloodstream infections and infections in companion animals and poultry [38]. Originally identified in 2008, ST131 is associated with the worldwide spread of CTX-M-15 resistance and recent reports have also identified strains that are resistant to carbapenems [39]. Although the *bla*_CTX-M-15_ gene was not present in the *E. coli* ST131 strain from our study, other ESBL enzyme genes (*bla_OXA-1_*, *bla_SHV-12_*, *bla_TEM-1_*) as well as the metallo-beta-lactamase gene *bla_NDM-1_* were detected. Another ST, the *E. coli* ST155 strain, was also recently reported as a clonal group of animal origin that is spreading in humans and is highly drug resistant [40]. Our strain was carbapenemase-producer (OXA-48) and, in addition, resistant to aminoglycosides, tetracyclines and sulfonamides. Importantly, recent studies showed *E. coli* ST155 strains carrying the *mcr-1* gene and novel variant *mcr-1.26* [41,42]. However, our strain was susceptible to colistin in vitro and *mcr* genes were not detected. Moreover, some previous studies also detected *E. coli* ST155 strains in surface waters [43,44].

Occurrence of *Acinetobacter* spp. is well-described in the natural environment, with special attention to *A. baumannii*, as the most important species causing infections in humans [45]. Isolation of carbapenem-resistant *A. baumannii* (CRAB) from water environments was documented in multiple studies [46]. We detected two CRAB isolates. One isolate harboured *bla*_OXA-66_, chromosomaly located, inherited carbapenemase [47], and *bla*_OXA-23_. This isolate belonged to ST2, the most widespread clone globally [48]. ST2/*bla*_OXA-23_-positive *A. baumannii* was one of the most prevalent clones among CRAB hospital isolates from Serbia [49]. Furthermore, in the same study, the most frequent clone circulating in Serbia was ST492, *bla*_OXA-66_- and *bla*_OXA-72_-positive, the same as the second isolate from our research. According to Lukovic et al., a ST492/*bla*_OXA-72_-positive clone was rarely described, and only in isolates that originated from Serbian patients [49].

Finally, the most worrying findings of the study is the high level of concurrent resistance to last-resort antibiotics carbapenems and colistin in Gram-negative clinically relevant wastewater isolates in Serbia. The inappropriate use of colistin has been shown to provide selective pressure for the emergence of colistin and multidrug (including carbapenem) resistant strains. Interestingly, different mechanisms of chromosomal- and plasmid-mediated resistance to colistin were detected in isolates from the study. First, a missense mutation in the *mgrB* gene resulting in MgrB^C28S^ was identified in 6 of 7 (85.7%) ST101 *K. pneumoniae* isolates. Although other amino acid substitutions (C28F; C28Y) have been described at the same position of the MgrB regulator in *Klebsiella* spp. [50], the substitution C28S was first described by Palmieri et al. in ST101 *K. pneumoniae* human isolates [21]. To our knowledge, this is the second study that confirms the presence of this novel mutation in the *mgrB* gene of *K. pneumoniae* in Serbian isolates. A genetic change (mutation or deletion) in almost any position of *mgrB* predominantly leads to functional inactivation of the MgrB peptide, which cannot act as a feedback inhibitor of the PhoP/PhoQ system; this plays a significant role in colistin-resistant *K. pneumoniae* strains [50]. Second, one *E. cloacae* complex isolate harbored plasmid-mediated mobile colistin resistance gene *mcr-9.1*. *Enterobacter* spp. and *Salmonella* spp. are the main hosts of *mcr-9.1* gene globally [51], and can be found in different reservoirs (human, animal, food and environment) [52], making this resistance mechanism an issue under the perspective of One Health. This *mcr* variant was previously confirmed in a *K. pneumoniae* human isolate from Serbia [53], indicating potential interspecies spreading of plasmid-mediated colistin resistance. Third, mutations in the *basRS* genes were detected in one *K. pneumoniae* and two *P. aeruginosa* isolates. The PmrAB (also termed BasRS) two-component system plays a crucial role in mediating the modification of LPS which leads to colistin resistance in Gram-negative bacteria and mutation was previously detected in various *Enterobacterales*, *Pseudomonas* spp. and *Acinetobacter* spp. strains [54]. Therefore, we hypothesize that the high rates of colistin resistance observed in our study may be connected to the widespread use of colistin, particularly in hospital settings in Serbia.

The authors acknowledge the limitations of the study, mainly the reduced number of wastewater samples and bacterial isolates targeted. Future studies should also consider including samples from hospital wastewater and receiving water bodies in order to track the environmental dissemination of ARB. However, in countries with poor wastewater treatment like in Serbia, AMR findings in wastewater are mirrored in their recipients. Overall, our study implementing state-of-the-art techniques for in-depth ARB investigation can be considered a good starting point for implementing a wastewater-based molecular epidemiology survey model to reflect the behavior of carbapenem-resistant Gram-negative bacteria in the community. The study also highlighted the need to minimize the spread of ARBs within hospitals before reaching the environment. The software application EPISEQ^®^ CS used here for WGS data analysis could also help to track, prevent, contain and stop the spread of healthcare-associated infections and AMR.

## 4. Materials and Methods

### 4.1. Study Setting and Sampling

Water samples were collected from four main sewer outlets in Belgrade prior to discharge to the Sava and the Danube rivers: site S1, the biggest combined sewer outlet “Sajam”, which accounts for almost 40% of the total of Belgrade’s wastewater discharge; site S2, combined sewer outlet “Lasta” drains the southwestern part of the central area of Belgrade; site S3, the largest separate sewer outlet “Ušće”; and site S4, “Istovarište”, the second largest combined sewer outlet in Belgrade. Figure 5 and Table 3 show locations of the sampling sites and their characteristics. Besides municipal wastewater, untreated hospital wastewater from tertiary healthcare hospitals in Belgrade are discharged in sewer outlets S1, S3 and S4 (S1 and S4 receive input from multiple hospitals, S3 from one tertiary healthcare hospital and S2 has no hospital input).

Sampling took place on four occasions between May and August 2018, with minimum time interval of one month between two independent sampling visits to minimize the possible overlapping of carryover of the targeted ARB. Wastewater was aseptically sampled in sterile 1L glass bottles. The samples were transported to the laboratory at a temperature of 4 °C and were processed on the day of collection.

### 4.2. Cultivation and Identification of Target ARB

For each water sample, 50 mL was passed through a 0.45 μm filter (Millipore, Billerica, MA, USA). Filters were incubated for 24 h at 37 °C in 5 mL of BBL fluid thioglycollate medium (Becton Dickinson, NJ, USA). Screening for target ARB was performed by streaking one loopful of enriched culture onto CHROMID^®^ CARBA SMART (bioMérieux, Marcy-l’Étoile, France). All colonies with different morphologies were picked and subcultured onto Columbia agar with 5% sheep blood (bioMérieux) at 37 °C for 18 to 24 h. Isolates were subjected to identification by protein profiling using matrix-assisted laser desorption/ionisation time-of-flight mass spectrometry (MALDI-TOF/MS) using a VITEK^®^ MS mass spectrometer (bioMérieux) equipped with Myla software. All target ARB were purified on Columbia agar with 5% sheep blood and preserved in cryotubes at −80°C.

### 4.3. Antimicrobial Susceptibility Testing

The antimicrobial susceptibility profiles were performed by the VITEK^®^ 2 automated system (bioMérieux). Panels of following antimicrobials were used: amoxicillin-clavulanic acid (AMC), piperacillin- tazobactam (TZP), ceftazidime (CAZ), ceftriaxone (CTX), cefepime (CEF), ceftazidime-avibactam (CZA), ceftolozane-tazobactam (C/T), ertapenem (ERT), imipenem (IPM), meropenem (MEM), aztreonam (ATM), amikacin (AMK), gentamicin (GEN), ciprofloxacin (CIP), levofloxacin (LVX), chloramphenicol (CHL) and trimethoprim-sulfamethoxazole (STX). Susceptibility to colistin was determined by broth microdilution method (ComASP^TM^ Colistin, LiofilChem, Italy). Profiles were evaluated according to the EUCAST breakpoint Table 2022, with category “I” meaning “susceptible, increased exposure”. Additionally, for *P. aeruginosa* and *A. baumannii* isolates, interpretation of antimicrobial susceptibility was performed only for antimicrobial agents with EUCAST clinical breakpoint for these bacteria (https://www.eucast.org/, accessed on 9 November 2022).

### 4.4. DNA Extraction and Whole-Genome Sequencing

Bacterial DNA from the overnight cultures of the study isolates was extracted with a DNeasy UltraClean Microbial Kit (Qiagen, Hilden, Germany) according to the manufacturer’s instructions. Quantitation was performed with Qubit^®^ dsDNA BR Assay kit (Thermo Fisher Scientific, Waltham, MA, United States) on Invitrogen Qubit^®^ Fluorometer (Thermo Fisher Scientific). DNA quality was checked using the 260/280 ratio absorbance parameter as determined by the DS-11FX spectrophotometer (DeNovix, Wilmington, DE, USA). Size measurement of genomic DNA was pefromred with HS Genomic DNA 50 kb Kit (Agilent Technologies Inc., Santa Clara, CA, USA) on Agilent Fragment Analyzer system.

Libraries were prepared using the Nextera XT DNA Library Preparation Kit (Illumina, San Diego, CA, USA) and Index Kit v2 Set A (Illumina). Library quality controls were performed with High Sensitivity DNA kit (Agilent) on Agilent 2100 Bioanalyzer system, and with Ultra Sensitivity NGS kit (Agilent) on Agilent Femto Pulse system. Library pools were prepared according to the previously described equimolar pooling strategy for multiplexing multiple bacterial species [55]. Sequencing was performed using MiSeq^®^ Reagent kit v3 (600-cycle) on a MiSeq^®^ Platform (Illumina) to generate 2 × 200 base paired-end reads.

The sequencing data were analyzed with the easy-to-use, fully integrated web-based software application EPISEQ^®^ CS, version 1.2.0 (bioMérieux). The application was based on reference-free approach with automated workflow (https://www.biomerieux-episeq.com/cs-how-it-works, accessed on 9 November 2022). Sequencing data were quality checked and assembled, followed by quality control of assembled genomes. Species identity, initially selected by the user, and potential intra- and inter-species contamination were checked. Genomic strain characterization was performed through the generation of multi-locus sequence typing (MLST) results, serotyping and pathotyping results for *E. coli* samples, detection of virulence and antimicrobial resistance markers, and detection of plasmids. Allele calling was performed in a proprietary whole-genome MLST (wgMLST) scheme defined for each EPISEQ^®^ CS species. Finally, based on the wgMLST allelic profiles, an epidemiological analysis was calculated and visualized in a dendrogram (% similarity) and a minimum spanning tree (number of allelic differences). For isolates (*n* = 7) with undefined sequence type (BS11104; BS11124) or undetermined allele variant (BS11106; BS11109; BS11113; BS11117; BS11118) in EPISEQ^®^ CS MLST allelic profile, complementary results were obtained from WGS data (assembly FASTA file) using https://cge.food.dtu.dk/services/MLST/ (accessed on 9 November 2022) tool with a minimum depth for an allele set to only 5x. Genome sequences and allelic profiles from these isolates with unknown STs were submitted to PubMLST (*E. cloacae* complex and *K. oxytoca*) [56] or Pasteur (*K. pneumoniae* and *E. coli*) for the assignment of new STs by database curators.

### 4.5. Identification of Missense Mutation in mgrB Gene Resulting in MgrB^C28S^

For the colistin resistance mechanism via mutations in the chromosomal *mgrB* gene, regarding substitutions of the cysteine amino acid (AA) at position 28 in the MgrB regulator of *Klebsiella* spp., only C28F and C28Y were available in the EPISEQ^®^ CS version (1.2.0) used for the study. A novel substitution (C28S) associated with ST101 *Klebsiella pneumoniae* was recently described [21]. To search for missense mutation in *mgrB* causing the substitution C28S in MgrB peptide (47 AA residues), using Geneious Prime^®^ (2022.0.1) software the wild-type sequence of *mgrB* gene (with TGC codon encoding cysteine at position 28) was first extracted from annotated NCBI RefSeq chromosome assembly (Accession number: NZ_CP065838.1) of *Klebsiella quasipneumoniae* strain KqPF26, and then aligned against the genome assembly of all studied ST101 *Klebsiella pneumoniae* isolates (*n* = 7).

## 5. Conclusions

The results of this pilot study on the detection of clinically relevant strains of carbapenem-resistant Gram-negative bacteria in wastewater prior to outlet into the international rivers without previous treatment is alarming and appears an emerging future public-health issue which demands increased attention and the proposal of mitigation measures. Importantly, the plethora of determined co-existence of ARG and virulence genes in investigated ARBs highlighted the serious threats of these pathogens on human and animal health. Immediate actions as a consequence of our study results include the proper treatment of wastewater. We recommend the development of new strategies for treating hospital and municipal wastewater effluents, and the establishment of a surveillance system to monitor MDR clinically relevant bacterial species in main sewer outlets and surface waters, as it is proposed by the EU Commission for systematic surveillance of SARS-CoV-2 and its variants in EU wastewaters [57].

## Figures and Tables

**Figure 1 antibiotics-12-00350-f001:**
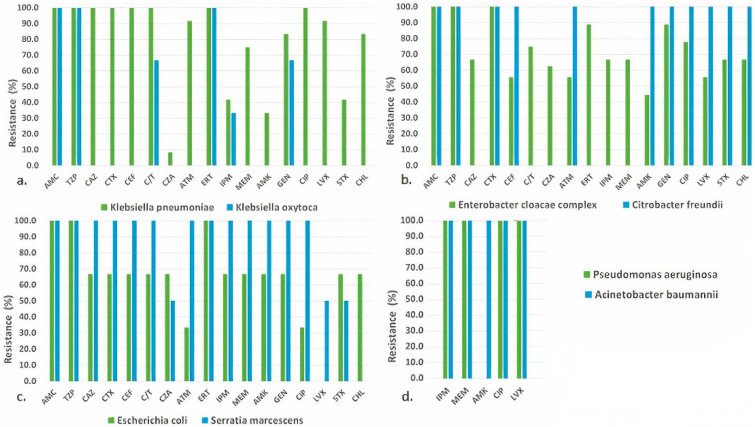
Resistance to antimicrobial agents among isolates of (**a**) *K. pneumoniae* and *K. oxytoca;* (**b**) *E. cloacae* complex and *C. freundii*; (**c**) *E. coli* and *S. marcescens*; (**d**) *P. aeruginosa* and *A. baumannii*. Abbreviations for antimicrobial agents: AMC, amoxicillin-clavulanic acid; TZP, piperacillin-tazobactam; CAZ, ceftazidime; CTX, ceftriaxone; CEF, cefepime; CZA, ceftazidime-avibactam; C/T, ceftolozane-tazobactam; ERT, ertapenem; IPM, imipenem; MEM, meropenem; ATM, aztreonam; AMK, amikacin; GEN, gentamicin; CIP, ciprofloxacin; LVX, levofloxacin; CHL, chloramphenicol and STX, trimethoprim-sulfamethoxazole.

**Figure 2 antibiotics-12-00350-f002:**
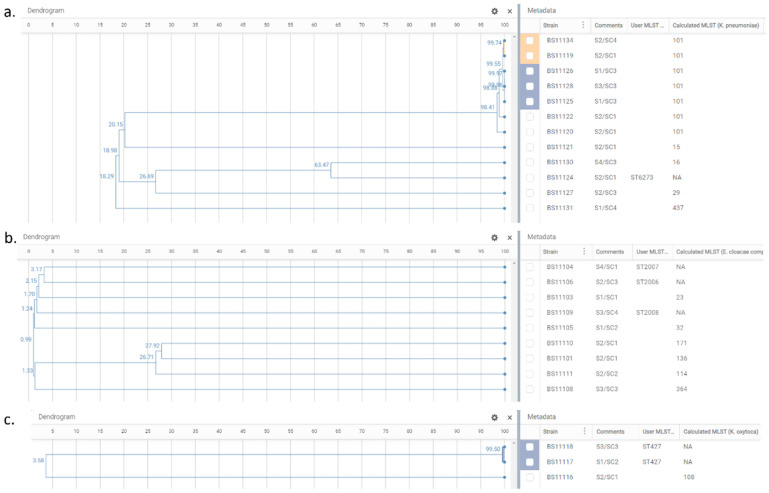
Phylogenetic trees of *K. pneumoniae* isolates, *n* = 12 (**a**); *E. cloacae* complex isolates, *n* = 9 (**b**); and *K. oxytoca* isolates, *n* = 3 (**c**). Dendrogram panel reflects the relationships between the samples based on their wgMLST profiles and shows the similarity percentage on nodes. Metadata panel includes information relative to sample identifier (Strain), sampling site/campaign (Comments), and MLST calculated by EPISEQ^®^ CS (Calculated MLST) or provided by the database curator when a new ST assignment was required (User MLST). The colored blocks show the clusters of closely related isolates detected by EPISEQ^®^ CS.

**Figure 3 antibiotics-12-00350-f003:**
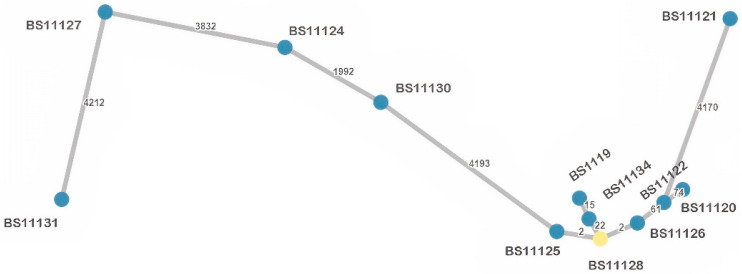
Minimum spanning tree (MST) based on wgMLST allelic profiles of the analyzed *K. pneumoniae* isolates (*n* = 12). Each dot represents one isolate with its identifier. The number of allelic differences between two isolates is shown on the connecting branch between them.

**Figure 4 antibiotics-12-00350-f004:**
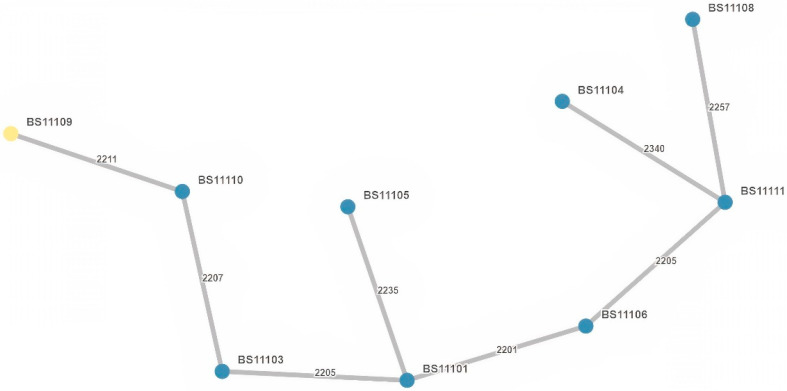
Minimum spanning tree (MST) based on wgMLST allelic profiles of the analyzed *E. cloacae* complex isolates (*n* = 9). Each dot represents one isolate with its identifier. The number of allelic differences between two isolates is shown on the connecting branch between them.

**Figure 5 antibiotics-12-00350-f005:**
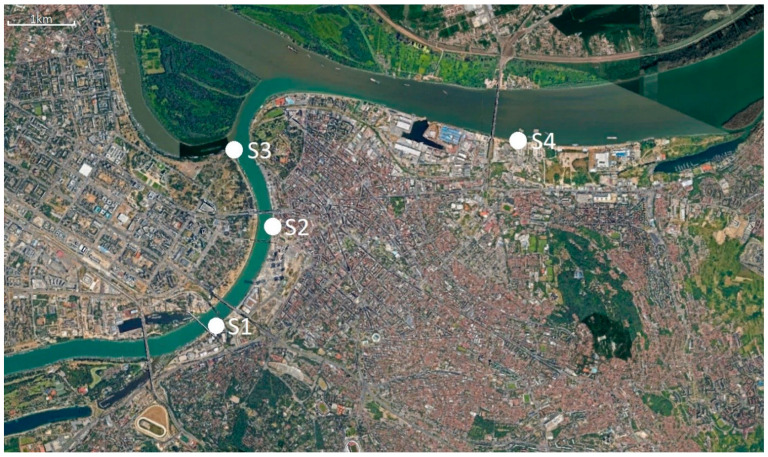
Satellite image of sampling sites (Google Earth).

**Table 1 antibiotics-12-00350-t001:** Distribution of isolated clinically relevant antibiotic-resistant Gram-negative bacteria.

S/SC	SC1	SC2	SC3	SC4
S1	*K. pneumoniae* (*n* = 1) *E. coli* (*n* = 1) *A. baumannii* (*n* = 1)	*K. pneumoniae* (*n* = 1) *K. oxytoca* (*n* = 1) *E. cloacae* (*n* = 1) *E. coli* (*n* = 1)	*K. pneumoniae* (*n* = 2)	*K. pneumoniae* (*n* = 1) *P. aeruginosa* (*n* = 1)
S2	*K. pneumoniae* (*n* = 2) *K. oxytoca* (*n* = 1) *E. cloacae* (*n* = 2) *S. marcescens* (*n* = 1)	*K. pneumoniae* (*n* = 1) *E. cloacae* (*n* = 1)	*K. pneumoniae* (*n* = 1) *E. cloacae* (*n* = 1) *E. coli* (*n* = 1)	*K. pneumoniae* (*n* = 1)
S3	/	/	*K. pneumoniae* (*n* = 1) *K. oxytoca* (*n* = 1) *E. cloacae* (*n* = 1)	*E. cloacae* (*n* = 1) *E. coli* (*n* = 1) *S. marcescens* (*n* = 1)
S4	*E. cloacae* (*n* = 1) *C. freundii* (*n* = 1)	*P. aeruginosa* (*n* = 1)	*K. pneumoniae* (*n* = 1) *A. baumannii* (*n* = 1)	/

S, sampling site; SC, sampling campaign; SC1, May 14, 2018; SC2, June 18, 2018; SC3, July 16, 2018; SC4, August 20, 2018; sampling sites are described in Section 4.1.

**Table 2 antibiotics-12-00350-t002:** Antibiotic resistance genes found in isolated target antibiotic-resistant bacteria.

Species	Number	ST	β-lactams	Aminoglycosides	Quinolones	Phenicol	Trim	Sulfo	Fosfo	Tet
*K. pneumoniae*	BS11119	101	*blaCTX-M-15*, *blaOXA-1*, *blaOXA-48*, *blaSHV-212*	*aac(3)-IIe*, *aac(6′)-Ib-cr*	*oqx*A, *oqx*A10, *oqx*A8, *oqx*B17	*cat*B3	*dfr*A14	/	*fos*A	*tet*(D)
*K. pneumoniae*	BS11120	101	*blaCMY-16*, *blaCMY-4*, *blaCTX-M-15*, *blaNDM-1*, *blaOXA-10*, *blaOXA-48*, *blaSHV-212*, *blaTEM-150*, *blaTEM-156*, *blaTEM-168*, *blaTEM-171*, *blaTEM-181*, *blaTEM-183*, *blaTEM-231*, *blaTEM-237*, *blaTEM-54*, *blaTEM-90*	*aadA*, *aadA2*, *aph(3″)-Ib*, *aph(3′)-VI*, *aph(6)-Id*, *armA*	*oqx*A, *oqx*A10, *oqx*A8, *oqx*B17	*cml*A5, *flo*R	*dfr*A12, *dfr*A14	*sul*1, *sul*2	*fos*A	*tet*(A)
*K. pneumoniae*	BS11121	15	*blaCTX-M-15*, *blaOXA-1*, *blaOXA-48*, *blaSHV-106*, *blaSHV-205*, *blaSHV-28*, *blaTEM-1*	*aac(6′)-Ib-cr*, *aph(3″)-Ib*, *aph(6)-Id*	*oqx*A, *oqx*B20, *oqx*B1	*cat*B3	*dfr*A14	/	*fos*A6	*tet*(A)
*K. pneumoniae*	BS11122	101	*blaCTX-M-15*, *blaOXA-1*, *blaOXA-320*, *blaOXA-48*, *blaOXA-534*, *blaSHV-212*	*aac(3)-IIe*, *aac(6′)-Ib-cr*	*oqx*A, *oqx*A10, *oqx*A8, *oqx*B17	*cat*B3	*dfr*A14	/	*fos*A	*tet*(D)
*K. pneumoniae*	BS11124	6273	*blaCTX-M-15*, *blaOXA-1*, *blaOXA-48*, *blaSHV-215*	*aac(3)-IIe*, *aac(6′)-Ib-cr*, *aph(3″)-Ib*, *aph(3′)-Ia*, *aph(6)-Id*	*oqx*A, *oqx*B19, *oqx*B24, *oqx*B25, *qnr*B1	*cat*B3	*dfr*A14	*sul*2	*fos*A6	*tet*(A)
*K. pneumoniae*	BS11125	101	*blaCTX-M-15*, *blaOXA-1*, *blaOXA-48*, *blaSHV-212*	*aac(3)-IIe*, *aac(6′)-Ib-cr*	*oqx*A, *oqx*A10, *oqx*A8, *oqx*B17	*cat*B3	*dfr*A14	/	*fos*A	*tet*(D)
*K. pneumoniae*	BS11126	101	*blaCTX-M-15*, *blaOXA-1*, *blaOXA-48*, *blaSHV-212*	*aac(3)-IIe*, *aac(6′)-Ib-cr*	*oqx*A, *oqx*A10, *oqx*A8, *oqx*B17	*cat*B3	*dfr*A14	/	*fos*A	*tet*(D)
*K. pneumoniae*	BS11127	29	*blaCTX-M-15*, *blaOXA-1*, *blaOXA-48*, *blaSHV-187*	*aac(3)-IIe*, *aac(6′)-Ib-cr*, *aph(3″)-Ib*, *aph(6)-Id*	/	*cat*B3, *flo*R	/	*sul*2	*fos*A6	*tet*(A)
*K. pneumoniae*	BS11128	101	*blaCTX-M-15*, *blaOXA-1*, *blaOXA-48*, *blaSHV-212*	*aac(3)-IIe*, *aac(6′)-Ib-cr*	*oqx*A, *oqx*A10, *oqx*A8, *oqx*B17	*cat*B3	*dfr*A14	/	*fos*A	*tet*(D)
*K. pneumoniae*	BS11130	16	*blaCTX-M-15*, *blaOXA-1*, *blaOXA-48*, *blaSHV-145*, *blaSHV-179*, *blaSHV-194*, *blaSHV-199*, *blaSHV-226*, *blaSHV-26*, *blaSHV-78*, *blaSHV-98*, *blaTEM-1*	*aac(6′)-Ib-cr*, *aadA2*	*oqx*A, *oqx*A10, *oqx*B32	*cat*B3	*dfr*A12	*sul*1	*fos*A5	*tet*(A)
*K. pneumoniae*	BS11131	437	*blaCTX-M-15*, *blaOXA-48*, *blaSHV-11*	*aadA2*, *aadA2*, *armA*	*oqx*A, *oqx*B,	*cat*B3	*dfr*A12	*sul*1	*fos*A6	/
*K. pneumoniae*	BS11134	101	*blaCTX-M-15*, *blaOXA-1*, *blaOXA-48*, *blaSHV-212*	*aac(3)-IIe*, *aac(6′)-Ib-cr*	*oqx*A, *oqx*A10, *oqx*A8, *oqx*B17	*cat*B3	*dfr*A14	/	*fos*A	*tet*(D)
*K. oxytoca*	BS11116	108	*blaOXA-48*, *blaOXY-1-1*	*aph(3′)-Ia*	*/*	*/*	*/*	/	*/*	*/*
*K. oxytoca*	BS11117	427	*blaGES-5*, *blaOXA-10*, *blaOXA-17*, *blaOXY-5-1*, *blaOXY-5-2*	*aac(6′)-Ib4*, *aadA11*, *aph(3″)-Ib*, *aph(3′)-VIa*, *aph(6)-Id*	*/*	*/*	*/*	*sul*1, *sul*2	*/*	*/*
*K. oxytoca*	BS11118	427	*blaGES-5*, *blaOXA-10*, *blaOXA-17*, *blaOXY-5-1*, *blaOXY-5-2*	*aac(6′)-Ib4*, *aadA11*, *aph(3″)-Ib*, *aph(3′)-VIa*, *aph(6)-Id*, *aph(6)-Id*, *aph(6)-Id*	*/*	*/*	*/*	*sul*1, *sul*2	*/*	*/*
*E. cloacae*	BS11101	136	*blaACT-46*, *blaACT-69*, *blaCTX-M-228*, *blaOXA-1*, *blaTEM-1*	*aac(3)-IIe*, *aac(6′)-Ib-cr*, *aadA*, *aph(3″)-Ib*, *aph(6)-Id*	*qnrB1*	*catA1*, *catB3*	*dfrA14*	*sul*2	*fosA*	*tet(A)*
*E. cloacae*	BS11103	23	*blaACT-68*	*aac(6′)-Ib-cr5*, *aadA*, *ant(2″)-Ia*, *aph(3″)-Ib*, *aph(6)-Id*	*qnrE1*	*/*	*/*	*sul*1	*fosA*	*/*
*E. cloacae*	BS11104	2007	*blaNDM-1*, *blaOXA-1*	*aac(3)-IIe*, *aac(6′)-Ib-cr*, *aadA2*, *aph(3″)-Ib*, *aph(3′)-VI*	*oqxB9*, *qacE*, *qnrB1*	*catA1*, *catB3*	*dfrA14*	*sul*1	*fosA*	*/*
*E. cloacae*	BS11105	32	*blaACT-52*	*aac(6′)-Ib4*, *aph(3″)-Ib*, *aph(3′)-VIa*, *aph(6)-Id*	*/*	*/*	*/*	*sul*1	*fosA*	*/*
*E. cloacae*	BS11106	2006	*blaMIR-7*, *blaOXA-1*, *blaSHV-2*	*aac(6′)-Ib-cr*, *aac(6′)-Ip*, *aadA*, *aph(2″)-IIa*	*/*	*catB3*	*dfrA14*	*sul*1	*/*	*/*
*E. cloacae*	BS11108	364	*blaCMH-4*, *blaKLUB-1*, *blaNDM-1*, *blaTEM-1*, *blaVEB-1*,	*aac(3)-IId aac(6′)-Ib-cr5*, *aadA2*, *ant(2″)-Ia*	*qnrVC4*	*cmlA5*	*dfrA12*, *dfrA14*	*sul*1	*fosA*	*/*
*E. cloacae*	BS11109	2008	*blaCMG*, *blaIMI-2*	*/*	*oqxB9*	*/*	*/*	/	*fosA*	*/*
*E. cloacae*	BS11110	171	*blaACT-45*, *blaCMY-4*, *blaCTX-M-15*, *blaNDM-1*, *blaOXA-1*, *blaOXA-10*, *blaTEM-1*	*aac(3)-IIe*, *aac(6′)-Ib-cr*, *aadA*, *aph(3″)-Ib*, *aph(3′)-VI*, *aph(6)-Id*, *armA*	*qnrB1*	*catA1*, *catB3*, *cmlA5*	*dfrA14*	*sul*1, *sul*2	*fosA*	*tet(A)*
*E. cloacae*	BS11111	114	*blaACT-72*, *blaCTX-M-15*, *blaNDM-1*, *blaOXA-1*, *blaTEM-1*	*aac(3)-IIe*, *aac(6′)-Ib-cr*, *aadA*, *aadA2*, *aph(3″)-Ib*, *aph(6)-Id*,	*qacE*, *qnrB1*	*catA1*, *catB3*	*dfrA12*, *dfrA14*	*sul*1, *sul*2	*fosA*	*tet(A)*
*E. coli*	BS11113	1133/1970 *	*blaEC-15*, *blaNDM-1*, *blaOXA-1*, *blaOXA-10*	*aac(3)-IIe*, *aac(6′)-Ib*, *aac(6′)-Ib-cr5*, *aac(6′)-Ib4*, *aadA*, *aph(3′)-Ia*, *aph(3′)-VI*	*qnrA6*	*catB3*, *cmlA5*	*dfrA14*	*sul*1	/	*/*
*E. coli*	BS11114	21/155 *	*blaEC-18*, *blaOXA-48*	*aph(3″)-Ib*, *aph(6)-Id*	*/*	*/*	*/*	*sul*2	/	*tet(B)*
*E. coli*	BS11115	43/131 *	*blaEC-5*, *blaNDM-1*, *blaOXA-1*, *blaSHV-12*, *blaTEM-1*	*aac(3)-IId*, *aac(3)-IIg*, *aac(6′)-Ib*, *aac(6′)-Ib-cr5*, *aac(6′)-IIc*, *aadA2*, *aph(3′)-Ia*	*/*	*catA1*, *catB3*	*dfrA12*	*sul*1, *sul*2	/	*tet(D)*
*C. freundii*	BS11100	112	*blaCMY-75*, *blaCTX-M-162*, *blaKLUB-1*, *blaTEM-1*	*aadA2*, *armA*	*/*		*dfrA12*	*sul*1	/	*/*
*S. marcescens*	BS11148	NA	*blaGES-5*, *blaSHV-2a*, *blaSRT-2*	*aac(6′)*, *aac(6′)-Ip*, *aph(2″)-IIa*	*/*	*catA1*	*/*	*/*	/	*tet(41)*
*S. marcescens*	BS11149	NA	*blaKLUB-1*, *blaNDM-1*, *blaSRT-2*, *blaTEM-1*	*aac(6′)*, *aadA2*, *armA*	*/*	*/*	*dfrA12*	*sul*1	/	*tet(41)*
*A. baumannii*	BS11098	492/425 **	*blaADC-100*, *blaOXA-66*, *blaOXA-72*	*aadA2*, *abaF*, *ant(3″)-IIa*, *aph(3″)-Ib*, *aph(3″)-Ib*, *aph(6)-Id*, *armA*	*/*	*/*	*dfrA12*	*sul*1, *sul*2	/	*tet(B)*
*A. baumannii*	BS11099	2/195 **	blaADC-73, blaOXA-23, blaOXA-66	*aac(3)-Ia*, *aadA*, *abaF*, *ant(3″)-IIa*, *aph(3′)-VIa*, *aph(6)-Id*, *armA*	/	*catA1*	*/*	*sul1*	/	tet(B)
*P. aeruginosa*	BS11135	348	*blaOXA-847*, *blaPDC-108*, *blaPDC-172*, *blaPDC-239*, *blaPDC-25*, *blaPDC-264*, *blaPDC-289*, *blaPDC-308*, *blaPDC-346*, *blaPDC-382*, *blaPDC-416*, *blaPDC-421*, *blaPDC-471*, *blaPDC-71*, *blaPER-1*	*aph(3″)-Ib*, *aph(3′)-IIb*, *aph(3′)-VIb*, *aph(6)-Id*	*crpP*	catB7	*/*	*/*	*fosA*	*/*
*P. aeruginosa*	BS11136	2305	*blaOXA-395*, *blaPDC-216*, *blaPDC-257*, *blaPDC-313*, *blaPDC-334*, *blaPDC-402*, *blaPDC-43*, *blaPDC-452*, *blaPDC-52*	*aph(3′)-IIb*	*crpP*	*catB7*	*/*	*/*	*fosA*	*/*

* Pasteur/Warwick MLST schemes; ** Pasteur/Oxford MLST schemes; NA-not available MLST scheme; Trim-Trimethoprim, Sulfo-Sulfonamides, Fosfo-Fosfomycin, Tet-Tetracycline.

**Table 3 antibiotics-12-00350-t003:** Sampling sites and characteristics of wastewater sewer outlets [18].

Sampling Site	Sewer Type	Average Sewage Flow Rate (m^3^/day)	WW Load, p.e.	Hospital WW	Discharge
S1	Separate and combined	94,250	537,000	MMA, CHC DM	The Sava River
S2	Combined	17,500	74,800	/	The Sava River
S3	Separate	47,820	268,000	CHC BK	The Sava River
S4	Combined	46,650	246,000	UCCS, CHC Z	The Danube River

S1, sewer outlet “Sajam”; S2, sewer outlet “Lasta”; S3, sewer outlet “Ušće”; S4, sewer outlet “Istovarište”; p.e.—population equivalent, 1 p.e. = 60 gBOD5/day; BOD5–5-day biochemical oxygen demand; WW, wastewater; MMA, Military Medical Academy, university hospital with 1100 beds; CHC DM, Clinical Hospital Centre “dr Dragisa Misovic”, university hospital with 400 beds; CHC BK, Clinical Hospital Centre Bezanijska Kosa, university hospital with 360 beds; UCCS, University Clinical Centre of Serbia, university hospital with 3000 beds; CHC Z, Clinical Hospital Centre Zvezdara, university hospital with 750 beds.

## Data Availability

The data that supports the findings of this study are available within the article and its supplementary material. Sequencing read data are deposited at the European Nucleotide Archive (https://www.ebi.ac.uk/ena/browser/home, accessed on 31 January 2023) under the project accession number PRJEB58827.

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
