# Peer review of "Whole-Genome Sequencing Snapshot of Clinically Relevant Carbapenem-Resistant Gram-Negative Bacteria from Wastewater in Serbia"

_antibiotics, 2023, doi:10.3390/antibiotics12020350_

Round 1

Reviewer 1 Report

Make Figure 1 clear. It's not visible.

Table 2. It should be B-lactams, not B-lactamase

Figures 2 and 3 are also not clear.

Reduce the discussion content and keep only what is appropriate.

Why is only short-read sequencing used? The hybrid approach seems better.

Reviewer 2 Report

The manuscript describes a survey of the occurrence and characteristics of carbapenem-resistant Gram-negative bacteria (CR-GNB) in  wastewater from Belgrade, Serbia, sampled from 4 sites taken at 4 separate times between May and August 2018. The study investigated samples from 4 sewer outlets prior to outflow into the Sava and Danube rivers. Thirty-four CR-GNB isolates were selected for antimicrobial susceptibility testing (AST) and whole genome sequencing (WGS). AST revealed that all isolates were multidrug-resistant. WGS showed that the isolates belonged to eight different species and 25 different sequence types, seven of which were novel. K. pneumoniae (blaCTX-M-15/blaOXA-48) with novel plasmid p101_srb was the most frequent isolate, detected at nearly all sampling sites. Genes encoding resistance to a wide variety of antibiotics were characterised. Chromosome-mediated acquired resistance to colistin was detected in K. pneumoniae (mutations 28 in mgrB and basRS) and P. aeruginosa (mutation in basRS), while plasmid-mediated resistance was confirmed in E. cloacae complex isolates (mcr-9.1 gene). The highest number of virulence genes (>300) was 30 recorded in P. aeruginosa isolates.

The manuscript is generally well written and the investigations appear to have been performed in an acceptable manner.

Specific Comments

Table 1 – give brief details of the sampling campaigns in the legend.

Line 23

define STs

Line 51 -meaning unclear – please reword
Still, implementation of restrictive police in antibiotic prescription 50 has not decreased AMR as expected.

Line 86
Belgrade significantly deteriorate
Belgrade significantly deteriorates

Line 108
An overview on the antimicrobial resistance
An overview of the antimicrobial resistance

Line 113
Similar finding was noticed Similar findings were found

Line 123
7 (58.3%) isolate 7 (58.3%) isolates

Line 144
Genes for SHV-212 type of enzyme were detected exclusively 144 in K. pneumoniae isolates that belonging to ST101 (58.3%). Genes for the SHV-212 type of enzyme were detected exclusively 144 in K. pneumoniae isolates that belong to ST101 (58.3%).

Line 147
all K. pneumoniae isolates harboured point mutation 147 on ompK36 locus,
all K. pneumoniae isolates harboured the point mutation 147 in the ompK36 locus,

Line 152
and other types of ESBLs founded in single isolates.--> and other types of ESBLs were found in single isolates.

Line 156
for blaNDM-1 carbapenemase gene, and one isolate harboured blaOXA-48 gene.
for the blaNDM-1 carbapenemase gene, and one isolate harboured the blaOXA-48 gene.

Line 159
other with blaTEM-1 and blaNDM-1 genes.
the other with blaTEM-1 and blaNDM-1 genes.

Line 160
different cephalosporinase from ADC and PDC families were confirmed. different cephalosporinases from ADC and PDC families were confirmed.

Line 168
Whole-genome sequencing (WGS) analysis showed a heterogeneous AME types from aac-, aad-, aph-, ant- and aba- families, including armA gene, which confers to a high level of aminoglycoside resistance. Whole-genome sequencing (WGS) analysis showed heterogeneous AME types from aac-, aad-, aph-, ant- and aba- families, including the armA gene, which confers a high level of aminoglycoside resistance.

Line 175
mediated by fosA gene, was detected in all K. pneumoniae and P. aeruginosa isolates, and in 88.9% E. cloacae complex.
mediated by the fosA 175 gene, was detected in all K. pneumoniae and P. aeruginosa isolates, and in 88.9% of E. cloacae complex isolates.

Line 185
22.2% E. cloacae complex isolates.
22.2% of E. cloacae complex isolates.

Line 200
86.7% ST101 200 isolates.
86.7% of ST101 200 isolates.

Line 203
Plasmids detected in E. cloacae complex have showed
Plasmids detected in E. cloacae complex showed

Line 206

Each E. coli isolate had unique plasmid profile Each E. coli isolate had a unique plasmid profile

Line 210
no virulence gene was detected.
no virulence genes were detected.

Line 212
The vast majority belonged to antiphagocytosis virulence genes
The vast majority were antiphagocytosis virulence genes

Line 218
K. pneumoniae isolates had quite similar distribution
K. pneumoniae isolates had a quite similar distribution

Line 226
of which one isolate presented new ST (ST6273).
of which one isolate represented a new ST (ST6273).

Line 234
and third from the same sampling
and a third from the same sampling

Line 241
Minimum spanning tree (MST) displaying
A minimum spanning tree (MST) displaying

Line 245
Minimum spanning tree (MST) displaying
A MST displaying

Line 248
while other two were new sequence type ST427
while the other two were a new sequence type, ST427

Line 267

Brunches branches

Line 278

Belgrade sewer system The Belgrade sewer system

Line 284

in Belgrade sewer system prior to outlet to recipients and thereafter AMR pollution of two international rivers in their lower drainage basin, providing important addition to the detailed insight on the extent of the issue on European level.

in the Belgrade sewer system prior to outlet to recipients and thereafter AMR pollution of two international rivers in their lower drainage basin, providing important additions to the detailed insight on the extent of the issue on an European level.

Line 287

according to WHO priority list, due to MDR profile according to the WHO priority list, due to their MDR profile

Line 298

blaOXA-1 is in a top 15 ARGs blaOXA-1 is in the top 15 ARGs

Line 300

and blaOXA-1 gene in vast majority and blaOXA-1 genes in the vast majority

Line 304

described in K. pneumoniae isolate from healthcare worker in Dutch described in a K. pneumoniae isolate from a  Dutch  healthcare worker

Line 308

consistent finding a consistent finding

Line 311

Genetic environment The genetic environment

Line 315

IncL plasmid was confirmed in 50% K. pneumoniae in this study, including one 315 ST15 and one ST101 isolates. The IncL plasmid was confirmed in 50% of K. pneumoniae in this study, including one 315 ST15 and one ST101 isolate

Line 319

conferring to MDR phenotype conferring a MDR phenotype

Line 340

may present worrisome public health issue. may present a worrisome public health issue.

Line 356

One of these isolates possess virulence One of these isolates possesses virulence

Line 364

in previous study in a previous study

Line 367

from Balkans, India and Vietnam from the  Balkans, India and Vietnam

Line 372

to specific clone to a specific clone

Line 378

but separate clade but a separate clade

Line 380

and point out the need for further investigation of routs and points out the need for further investigation of routes

Line 388

Although blaCTX-M-15 gene Although the blaCTX-M-15 gene

Line 399

described in natural environment with the special attention described in the natural environment with special attention

Line 403

Inhereted Inherited

Line 409

isolates originated from Serbian patients isolates that originated from Serbian patients

Line 422

that confirm presence of this novel mutation in mgrB gene of K. pneumoniae, again in Serbian isolates. that confirms the presence of this novel mutation in the mgrB gene of K. pneumoniae in Serbian isolates

Line 449

before to reach the environment before reaching the environment
